# Target Acquisition for Handheld Virtual Panels in VR

## ABSTRACT

The Handheld Virtual Panel (HVP) is the virtual panel attached to the non-dominant hand's controller in virtual reality (VR). The HVP is the go-to technique for enabling menus and toolboxes in VR devices. In this paper, we investigate target acquisition performance for the HVP as a function of four factors: target width, target distance, the direction of approach with respect to gravity, and the angle of approach. Our results show that all four factors have significant effects on user performance. Based on the results, we propose guidelines towards the ergonomic and performant design of the HVP interfaces.

## CCS CONCEPTS

• **Human-centered computing** → **Empirical studies in HCI**.

**ACM Reference Format:**

. 2019. Target Acquisition for Handheld Virtual Panels in VR. In *Proceedings of ACM Conference (Conference'19)*. ACM, New York, NY, USA, 5 pages. https://doi.org/10.1145/1122445.1122456

## 1 INTRODUCTION

With the increasing popularity of consumer virtual reality (VR), we see more and more VR apps for creativity and productivity. These apps fundamentally require menus and toolboxes for the assortment of options and controls they offer. And the interaction artifact that is quickly becoming the go-to technique for this is the handheld virtual panel (HVP). The HVP provides the primary toolbox in Google's TiltBrush [15] (Figure 1(left)) and Blocks [14], Oculus's Quill [11] and Medium [10] (Figure 1(right)), and HTC Vive's MakeVR [18]. Szalvari et al. in 1997 [30, 31] proposed the personal interaction panel where the user hold a tracked tablet in the second hand while doing their primary interaction with the dominant hand using a stylus. HVPs extend that concept for virtual panels anchored to the controller in the non-dominant hand and using ray-tracing instead of a stylus. There are multiple advantages to such an interaction [20]. First, handheld windows move along with the user, so they are always within reach. Second, they do not overly clutter the user's view, unless explicitly moved by the user. Third, handheld windows take advantage of the proprioceptive sense because they are attached to the non-dominant hand.

However, even with the ubiquity of HVP in products and research literature, we do not have a sense of what factors govern performance of target selection in HVPs. Consequently, there is

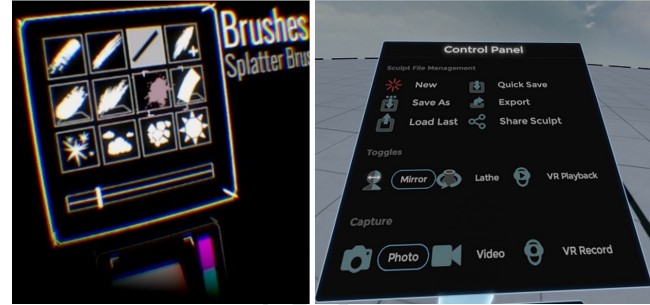

**Figure 1: (left) A handheld virtual panel (HVP) in Google Tilt Brush. (right) A HVP in Oculus Medium. These are used with a controller.**

a need to understand and quantify HVP target selection performance while considering these two factors: 1) hand motion here is governed by the direction of motion in relation to the ground due to the effects of gravity, and (2) since both the target and the pointer can be moved and controlled by the user during acquisition, user's approach will vary depending on the angle of movement in addition to distance and width.

We conduct a study to measure HVP target acquisition performance in relation to four factors that relate to the direction of movement with respect to gravity, the angle of movement with respect to the body, distance, and width. The results show that the performance depends significantly on all four factors. Based on the results, we propose guidelines towards the ergonomic design of the HVP interfaces.

## 2 RELATED WORK

### 2.1 Handheld Virtual Panels

In 1993, Feiner et al. [12] described three types of 2D windows in a virtual or augmented environment: Surround-fixed that are displayed at a fixed position within the surrounding, Display-fixed that are fixed at a position relative to the display itself, and World-fixed (or Object-fixed) that are fixed to objects in the 3D world. The HVP is an instance of the object-fixed window with the object being the non-dominant hand. Before Szalvari et al.'s proposal of the personal interaction panel [31], other works proposed handheld panels for specific VR scenarios using pen-and-tablet interfaces where the non-dominant hand held a tablet and the dominant hand held a pen to tap or draw on the tablet [3, 5, 13, 29].

For instance, Stoakley et al.'s Windows-in-Miniature (WIM) [29] proposes a miniature copy of the virtual world in the non-dominant hand for navigation and manipulation. Other works study effects of visual [3, 17], haptic [20] feedback for bimanual input in VR with a panel in the non-dominant hand. Lindeman et al. [20] found that users are 21% faster in shape selection tasks when using handheld 2D panels similar to HVP compared to surround-fixed panels that

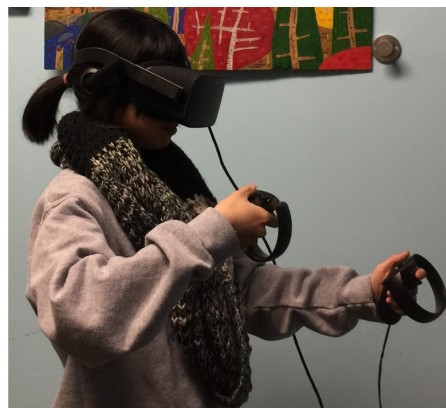

**Figure 2: A participant doing the study with the Oculus Rift and the two controllers.**

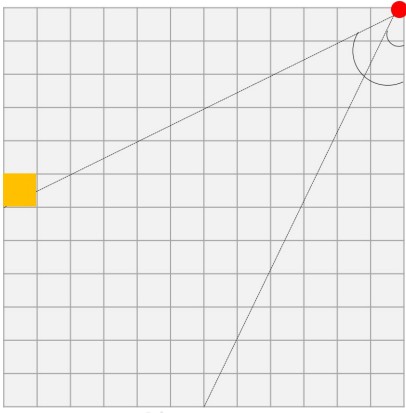

**Figure 3: The 12x12 (largest width) HVP schematic that the user sees in VR. Red dot denotes the pointer at one of its starting positions (other is bottom-right corner). The two angles 22.5° and 67.5° are denoted relative to the right edge. The yellow square shows a target to be selected. It is currently at 67.5° at maximum distance.**

float in front of the user. Similarly, Mine et al. [22] found that pointing to a target on a handheld panel was doubly fast than a fixed floating panel in space. However, none of the earlier works examine target acquisition in the HVP with respect to multiple target sizes and distances. Further, no existing work has examined the performance of the current HVP incarnation with the current hardware and interface. Consequently, we study the effect of distance and width on movement time for the HVP.

## 2.2 Whole-Handed 3D Movements in Air

While most works on handheld panels focus on direct pen or finger input, today's commercial VR systems rely on a controller in each hand with a ray tracing approach being used from the controller on the dominant hand to the targets on the panel. As hand tracking matures and becomes closer to commercial use in VR systems, we also hope to see explorations on hand-gesture based HVP interfaces. A related thread of work is ray tracing while using whole-handed 3D movements. Whole-handed 3D movements involve multiple limb movements, requiring higher muscular force and leading to variable movement trajectories, and hence variable pointing times [33]. Murata et al. [23] show that the direction of hand movement significantly affects movement time for a 3D pointing task. Following works [7, 34] found directional differences relating to shoulder and forearm motion. Zeng et al. [34] found that adduction movements are slower than abduction for 2D targets using hand motion in 3D space (detected by Kinect).

In our case, when using the HVP in current VR apps, a right-handed user holding the controller in the right hand usually approaches a tool on the panel in the left-hand from right to left direction. We investigate varying origins and angles in our study. There are other techniques and studies on target acquisition in 3D and in VR [1, 4, 8, 9, 21, 24, 25, 28, 32], but they address non-handheld, non-2D panel scenarios such as 3D object selection in the scene.

## 3 TARGET ACQUISITION STUDY

Aside from the traditional factors of distance and width, we need to take into account the effect of gravity for multiple starting positions and angles of movement.

### 3.1 Experiment Design

Figure 2 shows a participant doing the study. Similar to current HVPs, the dominant-hand controller raycasts a pointer into the scene. Figure 3 shows the HVP schematic that the user sees in VR. For selection, the user navigates the pointer on to the desired target and presses a button on the controller. The user can also move the non-dominant hand to move the target on the panel closer to the pointer. We investigated four independent variables: 1) STARTPOS: starting position of the pointer that determines the direction of movement with respect to gravity. STARTPOS has two levels, *top*: top-right and *bottom*: bottom-right position of the panel. 2) ANGLE: angle of movement relative to the right edge of the panel at StartPos that offers an additional level of nuance into the effect of gravity based on the angle of motion with respect to the gravity vector. It has two levels: 22.5° & 67.5°. Figure 3 shows the angles for the top StartPos. 3) DISTANCE: target distance from StartPos along the line of one of two angles. It has three exponentially increasing levels: 2cm, 6cm, 18cm. 4) WIDTH: target width. We keep the panel size constant and vary width by changing number of targets (all square shaped). Distance had three levels: 0.63cm (48X48 layout), 1.25cm (24x24), 2.5cm (12x12). Figure 3 shows the 12x12 layout. The panel size was kept slightly larger than existing panels in commercial applications to allow testing the distance factor with a larger range.

In total, there were 2x2x3x3=36 conditions and a within-subjects design was used. For each condition, participants performed 6 repetitions, resulting in 36x6=216 trials per participant. Owing to the large number of conditions, complete Latin square counterbalancing

across participants is not possible. Width was completely counterbalanced across participants. For each width, StartPos was completely counterbalanced across participants. For each width and startpos, the order of trials (consisting of Distance-Angle combinations) was randomized.

## 3.2 Participants

Twelve (7 female, 5 male) participants took part in the study (Range: 18-29, M = 22, SD = 3.004). All participants were right-handed and did not have any experience with VR. We believe the results will be similar for a mirrored study for left-handed users.

## 3.3 Apparatus and Task

The experimental application was developed in Unity3D. Participants wore an Oculus Rift head-mounted display and held Oculus Rift Touch Controllers, one on each hand, to interact with the VR environment. The task involved participants selecting targets on a HVP that is attached to the non-dominant hand, using the controller on the dominant hand that controls the raycast pointer. The user selects a target by clicking a button on the controller. For each trial, we measured the target acquisition time (time taken from the highlighting of the desired target until the button click), and errors (number of incorrect selections).

## 3.4 Procedure

After getting familiar with the apparatus and interface, participants performed 6 practice trials followed by the study. Before every trial, participants were required to bring the pointer back to the StartPos. The next target to be selected was highlighted 0.5s after the pointer was back at StartPos. Participants selected targets by bringing the raycasted pointer within the target's area (upon which a dark border indicated visual feedback), and pushing down on the trigger located at the back of their controller. We purposely avoided fatigue by mandating a 30s break after every 18 trials which the participants could extend if they wanted to. Upon incorrect selection, participants were not asked to redo the trial, but were given visual feedback that the selection was incorrect. Only the correct trials were part of the time analysis. Participants were instructed to perform the task as quickly and accurately as possible. At the end, a semi-structured interview was conducted.

## 4 RESULTS

### 4.1 Target Acquisition Time

We conducted a 4-way ANOVA and found main effects of all four variables on target acquisition time. However, there were interaction effects of StartPos*Distance ($F(1.224, 13.463) = 6.028, p < .05, \eta^2 = .354$ with *Greenhouse-Geisser correction (GG)*) and of StartPos*Angle($F(1, 11) = 21.776, p < .005, \eta^2 = .664$). Therefore, we ignore the main effects of StartPos, Angle, and Distance, and analyze the interaction effects. Since there were no interaction effects involving Width, we consider the main effect of Width ($F(2, 22) = 104.241, p < .001, \eta^2 = .905$). All posthoc tests described below have been conducted using Bonferroni correction.

*4.1.1 Effect of Width.* We conduct posthoc tests for Width, which show that the target acquisition time for all three widths is significantly different from each other ($p < 0.001$ for all). Figure 4(left) shows the effect of width on target acquisition time. Thus, the effect of Width is not affected by the other variables even though the other variables also have significant effects on time.

*4.1.2 Effect of StartPos and Distance.* The effect of Distance and StartPos have an interaction. We conducted 1-way ANOVAs for each of the two StartPos, *top* and *bottom* separately to see how distance affects the time in both. Figure 4(middle) shows the interaction plot. The effect of Distance is significant for both *top* ($F(2, 22) = 6.856, p < .01, \eta^2 = .384$) and *bottom* ($F(1.142, 12.558) = 23.728, p < .001, \eta^2 = .683$ with *GG*). For *top*, posthoc tests show the medium distance targets take significantly lower time than for small (p<.05) and large distance targets ($p < .01$). However, for *bottom*, both small and medium distance targets take significantly lower time than the large distance targets ($p < 0.01, p < .001$ respectively).

While the large distance targets expectedly perform worst, for *top*, the medium distance's performance is significantly lower. This is an interesting result and is possibly due to selective tilting of the controller by the participants depending on the target location. Participants use a combination of hand movement and orienting the controller in the hand to have the raycast pointer reach the target. Since the medium distance targets are in the middle of the panel, users can reach it with a combination of orientation change and hand motion. However, since even a slight orientation can result in a large displacement of the raycast pointer, smaller targets would be overshot with a flick of the wrist. With *bottom*, since the user is moving against gravity, the small and medium distances are comparable, but very much lower than the large distance targets.

*4.1.3 Effect of StartPos and Angle.* The effect of angle also depends on StartPos. Figure 4(right) shows the interaction plot. For *top*, $22.5°$ take a significantly lower time than $67.5°$ ($F(1, 11) = 11.793, p < .01, \eta^2 = .517$). For *bottom*, the inverse is true with $67.5°$ taking a significantly lower time than $22.5°$ ($F(1, 11) = 16.201, p < .005, \eta^2 = .596$). Again, owing to gravity, for *bottom*, the $22.5°$ angle requires the user to make more of an effort against gravity than $67.5°$. It's vice versa for *top* for the same reason.

### 4.2 Errors

No variable had a significant effect on error rate. While error rate values decreased with width (6.5%, 3.6%, 1.8%), the differences were not significant.

### 4.3 Qualitative Feedback

Unsurprisingly, majority of the participants reported the *bottom* starting position to be much more fatiguing. Some participants also mentioned that they thought that distance or angle had a very small effect on the difficulty of the task.

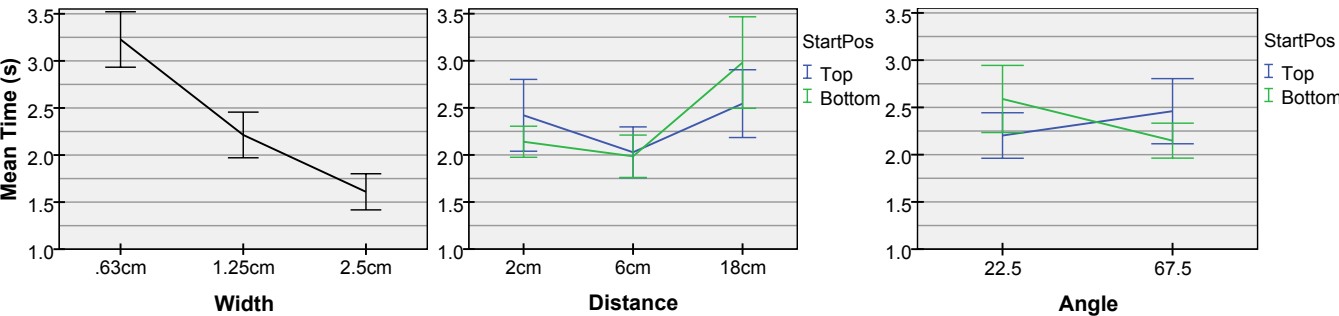

Figure 4: Target Acquisition Time results from the evaluation. We show only the main effects and the interaction effects. (left) Mean Time vs Width. (middle) Mean Time vs Distance for both StartPos. (right) Mean Time vs Angle for both StartPos. Error bars are 95% C.I.

## 5 DISCUSSION

### 5.1 Design Takeaways

The results suggest that gravity played a major part even when our experiment design minimized fatigue between conditions. The effect would be much more pronounced with longer, fatigue-inducing tasks. Most current HVPs use a cube-style panel with equal vertical and horizontal sizes. One simple solution to minimize the effect of gravity would be to have HVPs that have larger horizontal widths than vertical.

Our distance-based results suggest that minimizing hand motion and instead relying on wrist flicks to move the raycast pointer could help performance (see [26, 27]). Therefore, as opposed to having smaller panels, panel sizes can be increased (at least horizontally) to encourage the use of coarse wrist flicking.

Further, the design needs to minimize motion when the user is performing tasks below the panel (for instance, creating a ground texture) and will need to go against gravity to reach the HVP. One solution here would be arrange targets on the panel such that the high frequency targets are placed at the bottom of the panel, thus making them easier to reach from the bottom, while not overtly affecting the performance from top. Another possibility is to retarget the HVP [2] at a lower position while the non-dominant hand remains at the same position so that the user has to move less against gravity to reach the HVP. Retargeting has not been explored in the context of HVPs and could be a really useful technique to counter such effects. However, the tradeoff of increasing the visuo-haptic disconnect in this case would need to be explored.

Overall, we suggest three takeaways that should be considered by designers for HVPs depending on the context: 1) Panels with large horizontal widths as opposed to square shaped ones should be considered to counter effects of gravity and encourage wrist flicking, 2) Place high-frequency targets at the bottom of the panel, and 3) investigate retargeting of the HVP given the same non-dominant hand positions to minimize user motion against gravity.

### 5.2 Bimanual Parallel Input

While our work indicates some concrete directions to better the design of HVPs, one aspect that we did not explore in detail is the potential for HVPs to support bimanual parallel input. HVP is based on Guiard's kinematic chain model [16] for bimanual input, which proposes principles of asymmetric two-handed interface design. However, bimanual input may not always be useful. Buxton et al. [6] investigate parallelism, i.e., the degree to which the two hands are working parallelly, and concluded that participants are capable of parallelism and this improves task performance, but its use and efficiency depends on the mechanics of the task. Kabbash et al. [19] further showed that if a 2D task followed Guiard's model, it improves performance, and not following the model can worsen bimanual performance. With the HVP, users can potentially parallelly move both hands according to Guiard's kinematic chain model and improve their speed and performance. In addition to retargeting, bimanual parallel input is a promising direction for future exploration.

## 6 CONCLUSION

The handheld virtual panel is the most popular technique for accessing tools or menus in commercial VR creativity and productivity applications. In this paper, we conduct an evaluation of the target acquisition performance in the HVP as a measure of four variables. Our results show that all four have an effect on user performance. While there are expected effects such as reducing acquisition time with increasing width, the evaluation also suggests that gravity may be a crucial issue even when fatigue is minimized. Based on the results, we list takeaways to help improve the design of HVPs and indicate paths for future explorations. We believe addressing the limitations of HVPs uncovered in our study will go a long way in improving the user experience of HVP-based VR applications.

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
