# OpenReview forum: "Target Acquisition for Handheld Virtual Panels in VR"
_graphicsinterface.org/Graphics_Interface/2020/Conference — Submitted to GI 2020_

### Official Review · AnonReviewer2 · 2020-01-08
**This short paper presents the results of an experiment that studies the effect of angle and gravity in target acquisition for handheld VR panels. Although the paper is well written and has several strong points, I have major concerns about the validity of its conclusions.**

**Confidence:** 4
**Rating:** 5

**Review:**

The paper studies target acquisition in VR environments when users select menus from handheld virtual panels controlled with the non-dominant hand. In such scenarios, pointing performance can be affected not only by the size and distance of the targets but also by the angle of the movement and gravity. The paper presents the result of an experiment that studies the interactions of these factors.

This is a short 4-page paper with the appropriate depth, related work, and size of contribution. The paper is very well-written. Unfortunately, the submission does not include any video, thus some aspects of the task are not easy to understand. Overall, the paper is interesting and parts of the work are well executed. However, as I have major concerns about the experimental design as well as about the validity of some of the paper's conclusions/interpretations, I hesitate to argue for acceptance.

My main concerns are as follows:

The use of both hands (asymmetric bimanual input) is a key aspect that makes this task interesting and worth studying further. This is also part of the motivation of the paper (see Introduction). Unfortunately, the experiment does not clarify how the bimanual dynamics of this task affect user performance. I acknowledge that the paper discusses this limitation (Section 5.2) but I am not convinced. The paper mentions that participants could "also move the non-dominant hand to move the target on the panel closer to the pointer." But no further information is provided. I would at least need to know about the strategy that users employed to perform the tasks, as this largely determines the interpretation of the findings. Did participants move their the non-dominant hand to facilitate pointing? Notice that movement gravities are inverse for the two hands, so if gravity is  the factor of interest, the aspects needs further investigation. Furthermore, the experimental design does not seem to control for the initial positioning of the non-dominant hand. I would expect some additional information about this issue.

The effects that the authors observe may be due to physiological constraints that are specific to the angles and movement directions that they tested. Gravity may be less important. To support their interpretation, I think that the authors need to explore a wider range of movement directions, such as ones that move from left-top (or left-bottom) to the right. I wonder why the authors did not test a circular pointing task with additional angles (which eliminates the need for the Top/Bottom conditions).

I am not sure if I fully understand the experimental design from the schematic in Figure 3 and the text description. In particular, the angles do not seem to have a clear common definition for TOP and BOTTOM starting positions. I am further perplexed about the interactions shown in Figure 4. I cannot explain them, and the paper does not provide any clear intuition. Although the gravity interpretation seems to hold if we examine TOP and BOTTOM separately, the comparison between BOTTOM and TOP does not support this interpretation. Why isn't TOP always faster? As I explained above, physiological constraints may better explain the observed results.

Targeting performance across the two angles may have been further affected by the rectangular shape of the targets. The experiment should have tested circular targets instead.

For the same reasons, I don't feel comfortable with the design takeaways (Section 5.1), as they are not clearly supported by the findings. For example, the paper mentions that "he design needs to minimize motion when the user is performing tasks below the panel." Again, how is this supported by the results?

These are additional comments about the analysis of the results:
- Don't reporting on main effects (because of interaction effects) is a weird non-justified decision. Even if these effects are not "statistically" significant, the paper needs to report on them. I would also expect some analysis of the authors' expectations or prediction before running the experiment. As I mentioned above, one could expect that TOP is faster than BOTTOM. I would also expect that smaller distances would result in faster times. Disregarding the analysis of these main effects is not justified.

- I recommend reporting precise p-values rather than inequalities of the form p < .05 or p < .01. See: https://www.tandfonline.com/doi/full/10.1080/00031305.2019.1583913#_i2

---

### Official Review · AnonReviewer5 · 2020-01-08
**Relevant topic, but the study design needs to be improved**

**Confidence:** 4
**Rating:** 4

**Review:**

Pros
As the authors clearly state, handheld virtual panels are becoming ubiquitous in VR applications, yet no other paper has study how users interact with them. So, there is a need for a user study that evaluates different selection conditions. And this paper is at the correct time to start a discussion about the design of these menus.

Clarity
The description of the independent variables is difficult to understand, and this affects the understanding of the participant's actions.  The paper will benefit from additional figures. For example, an image of the participant view during the experiment. Or a diagram that explains the angles and the arm movements using a human figure.

Cons
There are some problems with the experiment design. In specific with the control of the non-dominant hand movement. As far as I understand, participants were free to move both hands in any direction they wanted. Yet, the authors only require participants to bring the dominant hand to a start position, but not the non-dominant hand. If this is correct, then every try could be different, as the participants could have used different muscles. The authors explain some of their results by using the non-dominant hand-wrist movement, but they do not discuss how having the menu in different positions can affect the selection. See Paul Lubos' work on pointing and joint movement, e.g 10.1145/2983310.2985753.
Also, the authors need to better justify some of their decisions regarding the experiment design and data analysis. First, why Width is part of the Latin-square, instead of Angle. Usually, in pointing experiments, distance and width are random variables as their effect are well know on the user performance. Second, the authors removed the incorrect trials for time analysis, but they do not explain why.
Finally, in the whole-handed 3D movements in the air section, the authors mix previous literature about the virtual-hand selection and ray-casting selection. For example, 7 and 23 are Fitts’ Law extensions for 3D virtual-hand pointing. At the same time, there is previous work missing related to 3D ray-casting pointing, i.e. 10.1016/j.ijhcs.2010.05.001.

Summary
I think the topic is relevant and well-timed. However, I think the paper needs more work, both to make it easier to understand and to better explain some of the decision regarding the study design. My main concern is that the authors discuss the importance of the second-hand movement and position, yet they did not control it on the user study. Based on that, I don't think I can recommend it for acceptance.

---

### Official Review · AnonReviewer1 · 2020-01-10
**Overall good work**

**Confidence:** 3
**Rating:** 7

**Review:**

Summary:
In this paper a study is conducted to analyze the performance of target acquisition on handheld virtual panels based on four variables: the width of the targets, distance from start position to desired target, direction of dominant hand movement with respect to gravity, and the angle of approach from start point to target. The results of this study claim that all of the above variables do significantly impact the time taken to select a given target. Based on these findings, some suggestions for more functional and user-friendly panel designs are made.


Review:
Overall the paper is good – ergonomics in virtual reality is a worthwhile and interesting area of research, and the authors make some good suggestions for implementation. The related work section was concise and relevant, and for the most part got me up to speed on the technology relevant to the study.  The related work mentions other kinds of virtual panels such as surrounding fixed and display fixed. It would be nice if there were a control condition with one of these other panel types in order to a) see if the results are generalizable, and b) see if any of the effects observed are due to also having control of the panel with the non-dominant hand.

Confirmation that starting from the bottom and working against gravity is more fatiguing is an interesting result – my assumption was that it would be negligible in such a study since regardless of the direction travelled for completing the selection, the user still would need to return to the start point after each trial, effectively retracing the same line in the other direction. Can it be confirmed that this effect is actually due to the effects of gravity, and not due to the fact that the angle of motion itself relative to the body is simply more ergonomic? Trying this myself in midair, the rotation angle of the shoulder joint is completely different between these two types of motion, and my immediate assumption for an explanation would be biomechanical rather than due to external forces. Also, it was noted that the user could move the non-dominant hand as a way to move the target panel closer to the pointer. If gravity were so fatiguing I would expect the users to do this more often when their dominant hand was starting at the bottom for economy of motion – was this ever observed?

My critique for “angle” is similar – interesting, but are we sure this is because of gravity and not economy of joint rotation?

The suggested solution to minimize the effects of gravity by having HVPs with larger horizontal widths than vertical, with high frequency targets being placed along the bottom is a good one and is easily implementable. Regardless of whether the observed effects are actually due to gravity or not, this suggestion should still result in a more ergonomic panel, as long as users typically approach such panels from the bottom when they haven’t been instructed to do so.

Finally, the discovery of the existence of an ideal distance is very useful, but only if the typical starting point for a user in a non-laboratory setting is well-defined. In the real world would the user have some semi-constant resting point that they return to in between selections rather than just hovering above the previous selection until another one is to be made? If the latter then perhaps the ideal distance could be used for the spacing between tiles rather than where to place the most important tiles. It was found that larger width corresponds with a shorter time, but this was perhaps an obvious result. Since we cannot make tiles arbitrarily large, there doesn’t exist some ideal panel size other than making them as large as possible based on the amount of necessary tiles/size of the HVP itself.

---

### Meta-Review · Area_Chair1 · 2020-01-10

**Recommendation:** Reject
**Confidence:** 4

**Metareview:**

On the positive note, the reviewers see a study on the ergonomics of handheld virtual panels in VR applications as valuable. However, 2 of 3 Reviewers found the experimental design to be unclear and have issues, rendering the results unreliable. I would refer the authors to the reviews for details on how they can improve this work for re-submission.

---

### Decision · Program_Chairs · 2020-01-11

Reject